# Aplastic Anemia in Triple X Syndrome

**DOI:** 10.3390/children10010100

**Published:** 2023-01-03

**Authors:** Mohammed Aldarwish, Israa Alaithan, Fatimah Alawami

**Affiliations:** 1Pediatric Hematology and Oncology Unite, Pediatric Department, Qatif Central Hospital, Qatif 32654, Saudi Arabia; 2General Pediatric Department, Qatif Central Hospital, Qatif 32654, Saudi Arabia; 3School of Medicine and Surgery, Imam Abdrahman Bin Faisal University, Dammam 31441, Saudi Arabia

**Keywords:** triple X syndrome, aplastic anemia, sex chromosome aneuploidies, bone marrow failure

## Abstract

Triple X syndrome is the most common sex chromosome aneuploidies (SCA) in females. Still, it is underdiagnosed because patients are usually without clear dysmorphism, and the syndrome is not associated with any significant congenital anomalies. We are reporting a case of a 5-year-old girl who presented with aplastic anemia, confirmed by a bone marrow aspiration and biopsy. Her complete workup showed that she has three copies of chromosome X, which, given the diagnosis of triple X syndrome, requires a supportive treatment but not a bone marrow transplant. Few cases of aplastic anemia with sex chromosome abnormalities have been reported. We are reviewing the triple X syndrome in different aspects of the presentation.

## 1. Introduction

Sex chromosome aneuploidies (SCA) are defined as the presence of an abnormal number or function of sex chromosomes X or Y, which can be either lost or gained [1]. This group of chromosomal disorders encompasses several distinct karyotype groups, including XXY (Klinefelter), XYY, XXX (Trisomy X), XXYY syndrome, and XO (Turner syndrome).

Triple X syndrome trisomy X, or 47, XXX is characterized by the presence of an extra X chromosome. It is the most common SCA and chromosomal abnormality in females. Although the incidence of triple X syndrome is approximately 1 in 1000 live female births, less than 10% are clinically diagnosed due to mild phenotype characteristics and the absence of physical anomalies at birth. The etiology of triple X syndrome results from the inappropriate separation of the X chromosome. This improper separation occurs during cell division, known as nondisjunction, mainly derived from maternal origin during meiosis I. Characteristics of physical or facial findings are nonspecific and not common but may include tall stature (at or more than 75th percentile), below-average head circumference (less than 50th percentile), epicanthal folds, hypertelorism, clinodactyly, overlapping digits, and hypotonia. Genitourinary abnormalities such as renal dysplasia or premature ovarian failure (POF), congenital heart defects, seizure disorders, and gastrointestinal abnormalities such as constipation and abdominal pain are some medical problems that may associate with triple X syndrome. On the other hand, puberty onset, sexual development, and fertility are usually preserved. There are substantial variations regarding developmental characteristics in triple X syndrome. Still, children typically have greater rates of speech-language difficulties, especially verbal and motor skills abnormalities and cognitive deficits with learning disabilities. Psychologically, anxiety and mood disorders such as depression and adjustment disorders are commonly present in these patients.

Diagnosis of triple X syndrome can be made through amniocentesis or chorionic villi sampling (CVS) during the prenatal period. Most of the cases were diagnosed through a prenatal diagnosis workup for any abnormality detected antenatally due to increased use of genetic testing. Postnatally standard karyotype (chromosomal analysis) is usually used for diagnosis whenever requested for any medical condition. Triple X syndrome has a variety of physical and behavioral phenotypes, but it is often undiagnosed. Management mainly depends on the patient’s age and the severity of phenotype symptoms [1,2,3,4]. Here, we report a case of aplastic anemia in triple X syndrome present with a fever and pancytopenia and with the absence of characteristic features.

## 2. Case Presentation

The patient is a 5-year-old girl delivered at term by normal vaginal delivery after an uneventful pregnancy to a healthy mother with no neonatal complications; she is fully vaccinated according to the Saudi national vaccine schedule, and there was no family history of any medical or surgical diseases. Her developmental history was normal, and she met all the developmental milestones at the expected time. She was known to have uncontrolled bronchial asthma with frequent exacerbation; she was on Salbutamol and a Flixotide inhaler and not on regular follow-up. She was in her usual state of health until one month prior to her presentation on March 2018, when she started to have fever lasting for one week, which was responding to antipyretic. The fever was associated with throat pain and difficulty swallowing. The family sought medical advice in the hospital, and she was managed with antipyretic and oral antibiotic for three days then discharged home with an impression of acute tonsillitis. Two days later, she was still febrile with no improvement in her symptoms, and she started to become hypoactive with a decrease in her appetite and oral intake. The family went to another hospital, investigations were done, and her blood investigations showed pancytopenia. The patient was admitted for IV antibiotics and further workups. Bone marrow biopsy and aspiration were advised for the patient, but the family refused initially. Two days after admission, she developed respiratory distress and shifted to isolation in the pediatric intensive care unit with an impression of community-acquired pneumonia (CAP). She was managed with oxygen supplementation via a face mask with frequent nebulization, stayed for three days in the pediatric intensive care unit with an overall stable general condition, and showed improvement in her respiratory status. She shifted to the pediatric medical ward to complete the IV antibiotic course for seven days. Her Complete Blood Count (CBC) persistently showed pancytopenia without improvement in her CBC parameters. The patient was discharged home with good general health and given a follow-up after two weeks in the hematology clinic. In the follow-up, she looked pale and had a poor appetite, but there was no change in her activity. Her CBC at the time of follow-up was showing WBC: 2.6 K/µL, Hb: 8 gm/dL, Platelet: 79 × 10^3^/uL, and her parents preferred not to proceed for bone marrow aspiration and biopsy; then, the family was advised to look for another medical opinion. One week after the follow-up, she spiked a fever after she had contact with a febrile person, and, for that, she was evaluated in our emergency department at Qatif Central Hospital, Qatif, Eastern Province, KSA. Her physical examination showed normal vital signs and growth parameters in the 50th percentile for height and head circumference and the 25th percentile for weight with a good body built, and no dysmorphic feature was appreciated. She looked pale but not jaundiced or cyanosed. Physical examination showed congested tonsils without exudate with no lymphadenopathy, mild hepatomegaly, and ejection systolic murmur. Other parts of the examination were unremarkable. Her initial laboratory investigations are shown in the Table 1, Table 2, Table 3, Table 4 and Table 5.

After the family consented for bone marrow aspiration and biopsy, the result showed: marked hypocellularity of 25% with no megakaryocytes, granulocyte 41%, erythroid precursors 51%, lymphocytes 8%, no abnormal cellular collection seen, and no evidence of myelodysplastic syndrome or malignancy (Figure 1 and Figure 2). She was given the diagnosis of aplastic anemia. The patient did not require any blood products transfusion and remained stable throughout her hospital stay. Her karyotyping showed three copies of chromosome X in all metaphases examined. Chromosomal breakage was negative. Flowcytometry for PNH was negative, myelodysplastic syndrome/bone marrow failure FISH panel genetic testing revealed an uncertain variant in two genes (Table 6) that are known to be associated with bone marrow failure, but the clinical significance of this specific genetic change is unknown. The human leukocyte antigen (HLA) typing did not match any of her family members, and she was referred to a tertiary hospital to consider a haploidentical HLA or match-unrelated donor.

Granulocyte-colony stimulating factor (G-CSF) was initiated two to three times per week. Her ANC before G-CSF was 0.44 × 10^3^/uL, and there was a marked response in her ANC to 1.89 × 10^3^/uL. The patient was kept in regular follow-up in the hematology clinic for four years with an acceptable range of CBC parameters. She is currently off G-CSF and there are no major infections. Her WBC (2.9–6.9 × 10^3^/uL), ANC (0.69–3.7 with a mean 1.82 × 10^3^/uL), count-ranging Hb (11.5–13.9 gm/dL), MCV (85–93 fL), platelet (59–78 mean 67 × 10^3^).

## 3. Discussion

Triple X syndrome trisomy X, or 47, XXX is considered the most common aneuploidies in females [5]. It occurs in 1 in 1000 live female births [6]. Most individuals with triple X syndrome are diagnosed incidentally on prenatal genetic screening [7] because they live normal lives and rarely have medical or clinical manifestations. Significant facial dysmorphology or characteristic physical features are not commonly found in these patients. However, minor physical findings can be seen in some individuals [5].

The literature describes the neurological and developmental aspects of aneuploidies in general, as they might have elements of neurocognitive problems and are at increased risk for early developmental delays, especially in speech-language development and motor development related to hypotonia. Attention deficit hyperactivity disorder (ADHD) is present in 25–35% of cases [5]. Mean IQ may be 10 to 15 points lower than siblings, but typically falls well within the normal range [6]. Genitourinary abnormalities are the most common medical problems these patients encounter. Others include congenital heart defects, seizure disorders, EEG abnormalities, and gastrointestinal problems [5]. The triple X syndrome has a mild phenotype, if any, and is not uncommon to be diagnosed later in life for reasons seemingly unrelated to aneuploidy. As in our case, the patient had hematological disease presenting with pancytopenia and was diagnosed with hypoplastic/aplastic anemia based on her bone marrow biopsy. Upon reviewing the literature, we found one article describing two cases with aneuploidies that present with aplastic anemia, and both were females [2]. One of them had triple X syndrome, and her presentation was similar to our patient, including age. Although she is stable with hypoplastic anemia, she is at risk to progress to severe aplastic anemia or transformation. In correlation to our patient, there might be an association between triple x syndrome and aplastic anemia; still, detailed genetic study is needed to further approve the relation and exact pathophysiology because the patient’s trisomy will give similar findings in any chromosomal abnormality. It was found that 11.9% of patients with acquired aplastic anemia had chromosomal abnormalities. Trisomy was found to be the commonest abnormality [8]. Many case reports of trisomies reported to have aplastic anemia, such as trisomy [5,8] and 1q [9,10,11].

In addition to that, like other trisomies, fetal hemoglobin is increased. There is also an increase in congenital aplastic anemia and in erythroid hypoplasias, with heterocellular distribution and Gγ/Aγ ratio being fetal-like, which is found in our patient, who has a high HbF percentage reaching up to 19% [12].

Aplastic Anemia (AA) can be inherited or acquired; the bone marrow is characterized by hypoplasia or aplasia, a paucity of hematopoietic stem and progenitor cells (HSPCs), and pancytopenia in the absence of cancer cells infiltration or fibrosis.

Hematopoietic stem cells (HSCs) are multipotent cells that have the potential to differentiate into mature blood cells in the peripheral blood and tissues. Loss of HSCs results in a reduction in all three-cell linage production, which is the primary pathophysiology of this disease. AA is a rare condition, with a triphasic incidence at 2 to 5 years during childhood, at 20 to 25 years, and at more than 55 to 60 years during adulthood, representing inherited and acquired causes, respectively [13]. The clinical presentation of patients with AA is variable and depends on the severity of peripheral pancytopenia. There may be hemorrhagic manifestations such as ecchymosis and spontaneous bruising or bleeding secondary to thrombocytopenia, recurrent fungal and bacterial infections with no response to empirical antibiotics treatment secondary to neutropenia, and fatigue and pallor secondary to anemia. Furthermore, patients with inherited AA may present with characteristic physical findings that indicate congenital disorders. AA is diagnosed by the presence of pancytopenia in the peripheral blood, in addition to bone marrow hypocellularity without signs of dysplasia or cancerous infiltrates. The treatment methods are determined by many factors: the severity of the patient’s condition, the patient’s age, and the presence of a matched sibling donor. Although AA is considered a rare disease, delay in diagnosis and treatment may result in a life-threatening condition [14].

Aplastic anemia can occur as a congenital or acquired condition in the pediatric population. Hereditary forms of aplastic anemia often result from disorders of chromosome fragility such as Fanconi anemia or dyskeratosis congenital. Acquired idiopathic aplastic anemia has been reported with cytogenetic abnormalities of various types. Several reports described aplastic anemia in patients with trisomy 21. Patients with trisomy 8 and autosomal translocations may be predisposed to develop aplastic anemia [2]. In our patient, we found her to be a carrier for FA with FANCP/SLX4 and heterozygotes for mutations in FA genes, which do not appear to have an increased risk of cancer compared with the general population [15].

Although acquired AA and hypoplastic myelodysplatic syndrome (hMDS) are difficult to distinguish from each other due to overlapping, the cytological/morphological are the main differences and may be subtle due to severe hypocellularity in some cases. For that reason, it is important to evaluate carefully in the context of other findings [16,17]. In our patient, the morphology and cytogenetic do not match the definition of MDS, and it is under aplastic anemia, so a follow-up of the patient is important, as the morphology might change over time.

To our knowledge, the risk of aplastic anemia in sex chromosome aneuploidies is not clearly studied, and yet there is some discussion about the relation of X chromosome inactivation and its contribution to cytogenetic presentation. Based on this case, aplastic anemia could be a presenting diagnosis for a patient with an extra X chromosome, but the coincidence is still there.

## 4. Conclusions

Physicians should be aware of this unusual presentation in triple X syndrome, as this syndrome usually has normal phenotypic features and is discovered incidentally. Finally, more extensive studies should be conducted to study the association between aplastic anemia and triple X syndrome.

## Figures and Tables

**Figure 1 children-10-00100-f001:**
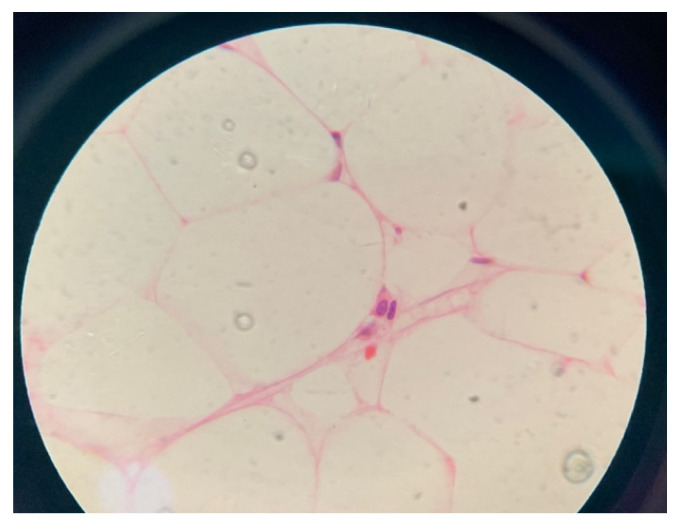
Microscopic picture of bone marrow tissue showing hypocellularity (average cellularity is 25%).

**Figure 2 children-10-00100-f002:**
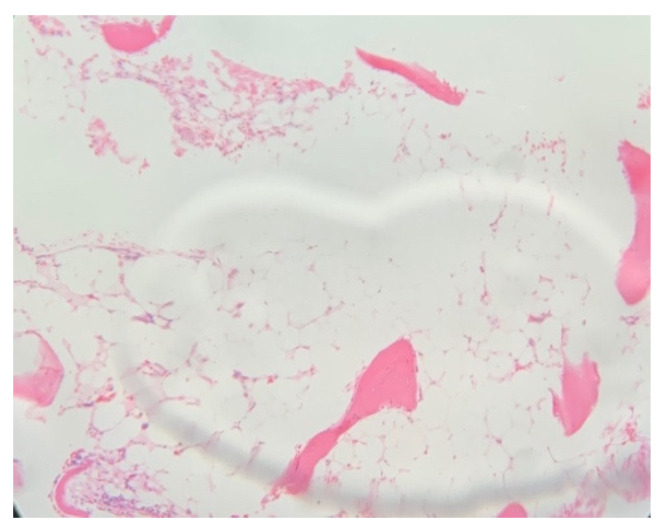
Microscopic picture of bone marrow tissue.

**Table 1 children-10-00100-t001:** Complete blood count (CBC).

Parameter	Value	Units	Reference Value
WBC ^1^	3.74	×10^3^	5–13
RBC ^2^	2.1	×10^6^/uL	4–5.2
Hb ^3^	6.3	g/dL	11.1–14.7
Hct ^4^	19.3	%	33–45
MCV ^5^	90.6	fL	76–90
MCH ^6^	29.6	PG	27–32
MCHC ^7^	32.6	g/dL	31.5–34.5
RDW ^8^	16.8	%	11.6–14
MPV ^9^	11.2	fL	7.4–10.9
Platelet count	79	×10^3^/uL	170–450
Neutrophil percentage	13.8	%	11.6–14
ANC ^10^	0.52	×10^3^/u	1.5–8
ARC ^11^	3.09	%	76–90

^1^ White blood cells, ^2^ red blood cells, ^3^ hemoglobin, ^4^ hematocrit, ^5^ mean corpuscular volume, ^6^ mean corpuscular hemoglobin, ^7^ mean corpuscular hemoglobin concentration, ^8^ red cell distribution width, ^9^ mean platelet volume, ^10^ absolute neutrophil count, ^11^ absolute reticulocytes count.

**Table 2 children-10-00100-t002:** Hb electrophoresis.

Parameters	Values	Units
Hb A	78.9	%
Hb A2	2.1	%
Hb F	19.0%	%

**Table 3 children-10-00100-t003:** Peripheral blood film.

Report
RBCs: dimorphic picture of normocytic and macrocytic cells, many teardrops poikilocytes, macro-ovalocytes, and no blast cells.

**Table 4 children-10-00100-t004:** Miscellaneous.

Parameters	Values
LFT ^1^, RFT ^2^, bone panel, and iron panel	Normal
Folate level, Vitamin B 12 level	Normal
Virology profile	Negative for hepatitis, CMV, EBN, and parvovirus.
Abdominal ultrasound	Hepatomegaly, spleen size appropriate for her age, no enlarged lymph node.

^1^ Liver function test, ^2^ renal function test.

**Table 5 children-10-00100-t005:** Marrow aspiration.

No megakaryocytes, granulocyte 41%, erythroid precursors 51%, lymphocytes 8%, no abnormal cellular collection seen, no evidence of myelodysplastic syndrome/malignancy.

**Table 6 children-10-00100-t006:** Bone Marrow Failure Syndromes Panel.

Gene Name	Variant	Interpretation
LYST (NM_000081.2) Inheritance: AR	c.8801 + 12G > C het	VUCS unable to predict
FANCP/SLX4 (NM_03244.2) Inheritance: AR	c.4921G > A (p. Va11641I1e) het	VUCS unable to predict

VUCS: Variant of uncertain clinical significance, AR: autosomal recessive, het: heterozygous. Gene of interest: AP3BI, BRCA2 (FANCD1), BRIP1 (FANCJ), CSF3R, CXCR4, DKCI, ELANE (ELA2), ERCC4, FANCA, FANCB, FANCC, FANCD2, FANCE, FANCF, FANCG, FANCI, FANCI, FANCM, G6PC3, GATA1, GATA2, GEI1, HAX1, LAMTOR (ROBLD3), LYST, MPL, NHP2 (NOLA2), NOP10 (NOLA3), PALB2 (FANCH), RAB27A, RAC2, RAD51C (FANCO), RBM8A, RMRP, RPL5, RPL11, RPLI5,RPL26, RPL35A, RPS7, RPS10, RPS17, RPS19, RPS24, RPS26, RTEL1, SBDS, SLC37A4, SLX4 (FANCP), SRP72, TAZ, TERC (hTR), TERT, TINE2, USB1 (c16orf57). VPS13B, VPS45, WAS, WIPF1, WRAP53 (TCABI, ADR79).

## Data Availability

Not applicable.

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
