# Peer review of "Aplastic Anemia in Triple X Syndrome"

_children, 2023, doi:10.3390/children10010100_

Round 1
Reviewer 1 Report
Dear Author,
This case report is really interesting. However, important informations are missing :
- what is the evolution from aplastic anemia diagnosis and this report? platelet nadir, hemoglobin nadir, does the patient require regular transfusion, has the patient developed any infection, is the patient eligible for immuno modulating treatment or stem cell transplantation? ...
- except for the finding of triple X syndrome, what were the results of the genetic work-up?
Otherwise, it would be interesting to gather the 2 reported cases of aneuploidie and aplastic anemia (but also the other case described in the paper of Rush et al) and your case in one table with the type of sex chromosome aneuploidie, the sex, the age at diagnosis of aneuploidie, the age at diagnosis of aplastic anemia, the results of the genetic work-up, the follow-up time.
Author Response
This patient might have rather MDS or undiagnosed constitutional bone marrow failure syndrome than AA.
I am basing my suspicion on high reticulocyte count, and high HbF. This doesn't fit into AA pattern.
The reported patient should undergo NGS testing.
Although acquired AA and hypoplastic myelodysplatic syndrome (hMDS) difficult to distinguish from each other due to overlapping each other. The cytological/morphological are the main differences and may be subtle due to severe hypocellularity in some cases and need to be evaluated carefully in the context of other findings. In our patient, the morphology and cytogenetic do not match the definition of MDS and it is under aplastic anemia, a follow-up of the patient is important as the morphology might be change with time.
fetal hemoglobin is increased, also, increased in congenital aplastic anemia and in erythroid hypoplasias, with heterocellular distribution and Gγ/Aγ ratio fetal-like which is found in our patient who has a high HbF percentage reaching 19%.
In our patient she was found to be a carrier for FA with FANCP/ SLX4 and heterozygotes for mutations in FA genes do not appear to have an increased risk of cancer compared with the general population.

Reviewer 2 Report
Review comment
The subject of the presented case report study was rare and interesting. I think that the fact that the findings are not very clear may contribute to the literature in making a faster diagnosis and intervention in these and similar patients.
Describing the case in detail, conducting the investigations in detail, and phasing out the intervention were strengths of the study. It is a well planned work, methodologically.
Apart from these strengths, there were some deficiencies in the writing system of the manuscript. The method part of the abstract is not clearly written. We expect readers to have a clear idea of the whole work when they read the summary.
The introduction part contained more descriptive information. We expect information on the results of other studies in the literature to be provided. A reference is given in the first sentence of the introduction and only at the end of a long paragraph. In article writing, it is more appropriate to give references in intermediate sentences instead of a block paragraph.
At the beginning of the discussion section, the information in the introduction section is given again. It is more appropriate to give this information at the entrance. In addition, the discussion section should not be an informative text such as an introduction. Discussion with the results of the research in the literature should be done to establish a relationship. The discussion should include more literature sources.
Minor revision
- -The method part should be added more clearly to the summary.
-More different sources should be added to the introduction and the sentences in between should include these sources.
- - A source containing the research results should be added to the introduction section.
- -The information in the introduction in the Discussion section should be transferred to the introduction, and the prevalences of "Attention Deficit and Hyperactivity Disorder" and "IQ" should be in the entry. The information put into the discussion must be discussed with the results.
